# Intra-lineage microevolution of *Wolbachia* leads to the emergence of new cytoplasmic incompatibility patterns

**Alice Namias***, **Annais Ngaku, Patrick Makoundou, Sandra Unal, Mathieu Sicard, Mylène Weill**\*

ISEM, Université de Montpellier, CNRS, IRD, EPHE, Montpellier, France

\* alice.e.namias@gmail.com (AN); mylene.weill@umontpellier.fr (MW)

## Abstract

Mosquitoes of the *Culex pipiens* complex are worldwide vectors of arbovirus, filarial nematodes, and avian malaria agents. In these hosts, the endosymbiotic bacteria *Wolbachia* induce cytoplasmic incompatibility (CI), i.e., reduced embryo viability in so-called incompatible crosses. *Wolbachia* infecting *Culex pipiens* (*w*Pip) cause CI patterns of unparalleled complexity, associated with the amplification and diversification of *cidA* and *cidB* genes, with up to 6 different gene copies described in a single *w*Pip genome. In *w*Pip, CI is thought to function as a toxin-antidote (TA) system where compatibility relies on having the right antidotes (CidA) in the female to bind and neutralize the male's toxins (CidB). By repeating crosses between *Culex* isofemale lines over a 17 years period, we documented the emergence of a new compatibility type in real time and linked it to a change in *cid* genes genotype. We showed that loss of specific *cidA* gene copies in some *w*Pip genomes results in a loss of compatibility. More precisely, we found that this lost antidote had an original sequence at its binding interface, corresponding to the original sequence at the toxin's binding interface. We showed that these original *cid* variants are recombinant, supporting a role for recombination rather than point mutations in rapid CI evolution. These results strongly support the TA model in natura, adding to all previous data acquired with transgenes expression.

## Introduction

*Wolbachia* are maternally transmitted endosymbiotic bacteria that infect up to 50% of arthropod species [1–3]. These bacteria are well known for their wide range of reproductive manipulations in arthropods. Their most common manipulation is cytoplasmic incompatibility (CI), i.e., in its simplest form, a reduction in embryo hatching rates (HRs) in crosses between infected males and uninfected females. CI results from a *Wolbachia*-induced modification that perturbs the first division of embryos, which infected females are able to rescue [4,5]. CI is thus formalized in a modification-rescue framework.

CI can also occur between 2 infected individuals if they are infected with genetically different and incompatible *Wolbachia* strains. In *Culex pipiens* mosquitoes, in which *Wolbachia*

deposited on GenBank (accession numbers are provided in S3 and S4 Tables).

**Funding:** This work was funded by the French MUSE project from the Université de Montpellier (reference ANR-16-IDEX-0006)(granted to MW). The funders had no role in study design, data collection and analysis, decision to publish, or preparation of the manuscript.

**Competing interests:** The authors have declared that no competing interests exist.

**Abbreviations:** CI, cytoplasmic incompatibility; GLM, generalized linear model; HR, hatching rate; LRT, likelihood ratio test; PCR, polymerase chain reaction; qPCR, quantitative polymerase chain reaction; SNP, single-nucleotide polymorphism; TA, toxin-antidote; *w*Pip, *Wolbachia* infecting *Culex pipiens*.

infection is fixed [6], the *Wolbachia* infecting *Culex pipiens* (*w*Pip) strains are responsible for a unique (to date) complexity of CI patterns based on multiple uni- and bidirectional incompatibilities [7–10].

In 2013, putative CI genes were identified using a combination of proteomic and genetic approaches [11]. Then, in 2017, two functional studies based on transgenic expression confirmed that these genes were key for CI, naming them *cif* for "CI factors" [12,13]. The *cif* genes were described in *cifA-cifB* tandems, playing a central role in CI, although the precise molecular mechanisms of CI are still to be determined. Although other models exist [14,15], in a growing range of *Wolbachia*, including *w*Pip, CI is thought to function as a toxin-antidote (TA) model [4,16–22]. In this model, a cross will be compatible if binding occurs between the *cidA*(s) (antidotes) present in the egg and the *cidB*(s) (toxins) present in the sperm, neutralizing *cidB*(s) toxicity [16,18,23,24].

Sequencing of *Wolbachia* genomes from different host species revealed several different pairs of *cif* genes, with distinct functional domains that categorized them into 5 clades [25–27]. All *w*Pip genomes sequenced so far have *cif* genes from 2 of these clades: clade I *cif*, in which the *cifB* gene has a deubiquitinase domain (the tandem is thus called *cid*, with *d* denoting the deubiquitinase domain), and clade IV *cif*, in which *cifB* bears a nuclease domain (thus called *cin*) [12,13]. Here, we use this functional-based nomenclature [28].

Before the discovery of the *cif* genes, models based on *Culex*-crossing experiments predicted that several factors or pairs of genes were required to encode the complex crossing patterns induced by *w*Pip [9,29]. Studies on *cif* (*cid* and *cin*) genes present in *w*Pip genomes highlighted that *cin* genes were always monomorphic, whereas *cid* genes were amplified and diversified in all sequenced *w*Pip genomes [21], with up to 6 different copies of each gene in a single *Wolbachia w*Pip genome. The different *cidA/cidB* copies in a given *w*Pip genome are known as "variants" and the full set of all copies constitutes the "*cid* repertoire." In *w*Pip genomes, the polymorphism of both *cidA* and *cidB* genes are located in 2 specific regions, which were named "upstream" and "downstream" regions and predicted to be involved in CidA-CidB interactions [21]. This has been confirmed by the recently obtained structure of a CidA-CidB cocrystal: out of the 3 interaction regions identified, 2 perfectly match the previously identified upstream and downstream regions for both CidA and CidB [23,24]. In addition to showing that CidA and CidB from *w*Pip bind together, this recent study also showed that the different CidA-CidB variants have different binding properties, thus lending further support to the TA model for explaining *w*Pip CI complexity [24].

Evolutionary changes of *Wolbachia* genomes on long time scales have been widely documented in different host species. The comparison of *cif* genes in *Wolbachia* strains from different arthropod hosts showed that they are quite divergent, and highly subject to lateral gene transfers [30,31] so that the congruence between *Wolbachia* and *cif* phylogenies is totally disrupted [25,26]. Such lateral transfers may be phage-linked, as *cif* genes are located in WO prophage regions [13,25], with lateral transfers of genes in prophage WO regions being previously documented [32–35]. Transposon-dependent transfers of *cif* genes have also been described [30]. By contrast, although the ability of *Wolbachia* to rapidly adapt to new environmental conditions has been suggested [36], very few studies have explored the short-term evolution of *Wolbachia*. To our knowledge, the only studies linking rapid changes in phenotypes (in a few host generations) to underlying genomic variations in *Wolbachia* were reported for the "Octomom region" in *w*Mel and *w*MelPop, showing that variations in the amplification level of this region were responsible for variations in both *Wolbachia* virulence and antiviral protection conferred by *w*Mel [37–39].

CI patterns in *Culex* were previously shown to change over a few host generations [40]. Yet, these changes were described before the discovery of *cif* genes. Here, we observed a change in the CI pattern between 2 isofemale lines kept in our laboratory since 2005: Slab (*w*Pip III) and

Istanbul (Ist, *w*Pip IV) [10]. While crosses between Slab females and Ist males were compatible from 2005 to 2017 [10,41], we observed fully incompatible crosses for the first time in 2021 (the reverse cross remaining incompatible). No changes in patterns were observed in the reverse cross (Slab males and Ist females), fully incompatible since 2005. This shift in CI patterns presented an opportunity to study the underlying genetic basis of CI evolution in laboratory-controlled isofemale lines.

The emergence of a new CI phenotype may have resulted from (i) contamination; (ii) the acquisition of a new toxin in some Ist *Wolbachia*; or (iii) the loss of antidote(s) in some Slab *Wolbachia*. We took advantage of a recent methodological development that enables the rapid and extensive acquisition of *cidA* and *cidB* repertoires using Nanopore Technologies sequencing [42] to address these 3 hypotheses. We were able to rule out the contamination hypothesis and show that no less than 3 distinct *Wolbachia* sublineages with different *cidA* repertoires evolved and now coexist in the Slab isofemale line (but do not appear to coinfect the same individuals). We also found, in accordance with TA model, that the loss of *cidA* variants (i.e., antidotes) in Slab females perfectly matches variations in their rescuing ability. The *cidA* variant whose presence/absence explains the change in compatibility has original amino acid combinations at its interaction interface with CidB, probably resulting from a recombination.

## Results

### From CI phenotype shift to underlying genomic basis

Slab and Ist isofemale lines are maintained at the lab since 2005 [10] and were crossed several times since: Slab females were compatible with Ist males at their establishment in 2005 (HRs above 70%, not precisely computed at that time) [10]. Crosses performed in 2017 (published in [41]) showed compatibility between Slab females and Ist males, but with highly variable hatching rates (HR ranging from 8% to 84% but all egg rafts produced larvae [41], S1 Fig). In the present study (crosses performed in 2021), we witnessed for the first time the emergence of fully incompatible crosses between some Slab females and Ist males. Twenty five rafts out of 94 had a 0% hatch rate (full CI), and the remaining 69 clutches had HRs between 3.2% and 96.4% (S1 Fig).

### At least 3 *cidA* repertoires coexist in the Slab isofemale line

To decipher the genomic changes that led to this recent CI shift, we sequenced the *cidA* and *cidB* repertoires present in *Wolbachia* infecting 4 individuals from the Ist isofemale line and 8 females from the Slab isofemale line (4 compatible and 4 incompatible) using Nanopore Technologies sequencing of *cidA* and *cidB* polymerase chain reaction (PCR) products [42]. While *Wolbachia* present in all Ist individuals had the same *cid* repertoire (presented in S1 Table), we found polymorphism in *cid* repertoires of Slab individuals: they all had the same *cidB* (i.e., toxins) diversity, composed of *cidB-III-ae3* and *cidB-III-ag1* variants, whereas *cidA* (i.e., antidotes) repertoires varied among individuals. *Wolbachia* present in compatible and incompatible Slab females all exhibited the variants *cidA-III-α(5)-25* and *cidA-III-γ(3)-12*, although additional variants were found in compatible females: *cidA-III-β(2)-25*, either alone or accompanied by *cidA-III-β(2)-16*. The observed differences in *cidA* repertoires can be summed up by the presence/absence of 2 specific *cidA* regions: the upstream region *cidA-III-β(2)* and the downstream region *cidA-III-16* (Figs 1 and 2A for nomenclature).

### *cidA* repertoire variation fully explains compatibility polymorphism

To determine if compatibility variations were linked to these specific *cidA* repertoire variations, we performed 3 replicates of the cross between Slab females and Ist males. We isolated

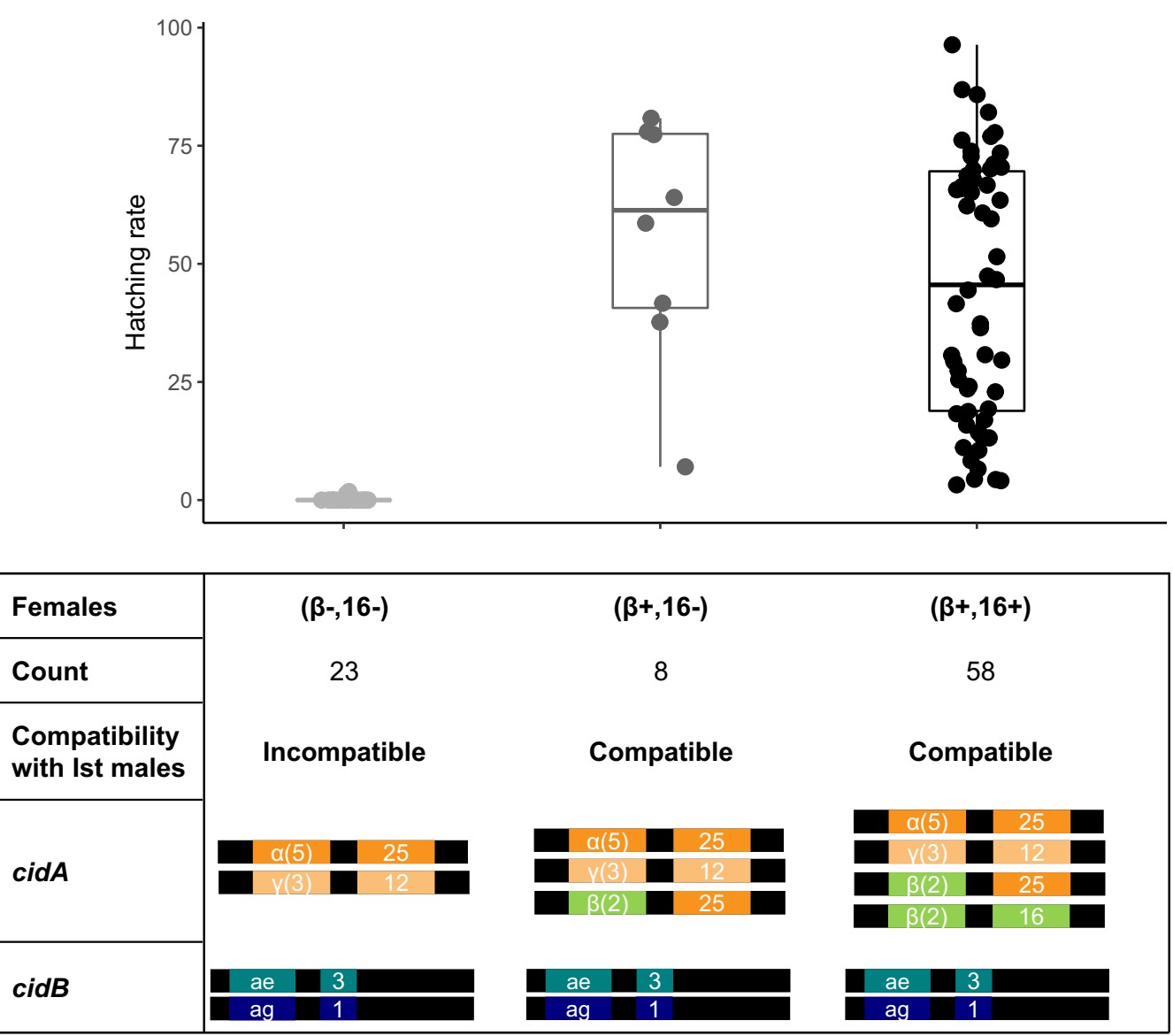

**Fig 1. Intra-line polymorphism in *cidA* repertoires matches the compatibility patterns.** Three distinct *cidA* repertoires coexist in the Slab isofemale line. Repertoires perfectly match with compatibility patterns, with females having repertoires (β+,16+) and (β+,16-) compatible with Ist males, while females (β-,16-) are incompatible with Ist males. The presence or absence of *cidA-III-β(2)* and *cidA-III-16* regions was determined by specific qPCRs. The presence of *cidA-III-β (2)* significantly influences HRs ($\chi$ = 13.094, $p < 0.0001$). Data supporting this figure are found in S1 Data. HR, hatching rate; qPCR, quantitative polymerase chain reaction.

egg rafts from a total of 102 Slab females from the same isofemale line (28, 34, and 40 females for replicates 1, 2, and 3, respectively) and studied their HRs (i.e., number of hatched larvae over the number of eggs laid). Eight females produced unfertilized rafts and were thus excluded from further analyses. For the remaining 94 females, HRs ranged from 0% (fully incompatible raft) to 96.4%, thus highlighting the strong differences in their rescuing abilities. Of these 94 females, 25 (i.e., 27%) were fully incompatible with Ist males representing the new emerged CI phenotype.

Since repertoires differed in the presence/absence of 2 specific regions of *cidA* repertoires, we designed primers and specifically screened the 94 mated females, of which 89 were

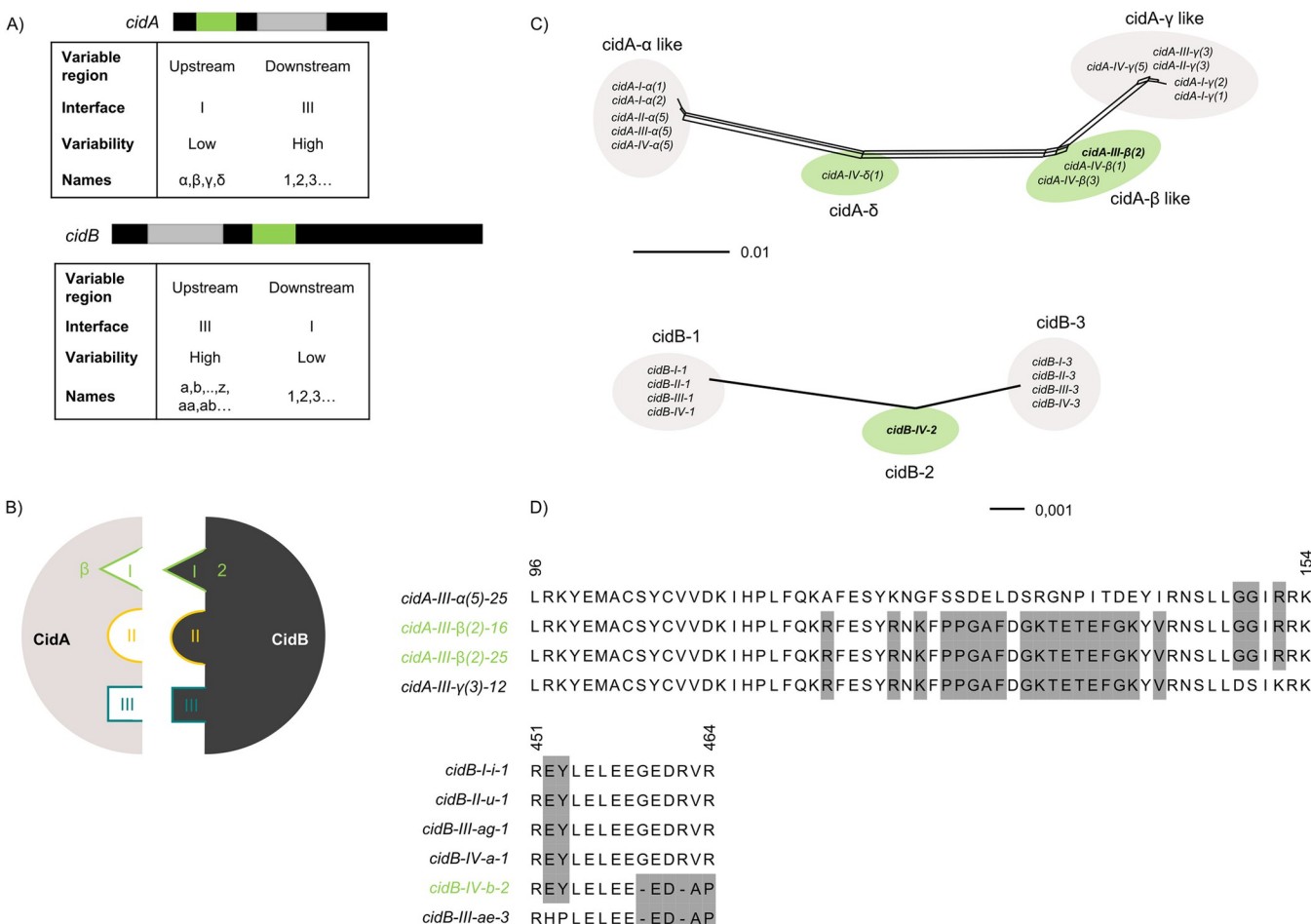

**Fig 2. Matching recombination in the CidA-CidB interaction interface I is key for rescue.** (A) *cid* gene polymorphism is located in 2 distinct regions for both *cidA* and *cidB*, named upstream and downstream regions. The upstream region of *cidA* and the downstream region of *cidB* show few variability, while the other 2 regions are highly variable. Variants names are composed of: (1) the name of the gene *cidA* or *cidB*, (2) the *w*Pip phylogenetic group concerned (I to V), (3) the name of the upstream region (in Greek or Latin letters for *cidA* and *cidB*, respectively), and (4) the name of the downstream region (in numbers for both genes). The name of a given upstream or downstream region corresponds to the same sequence across all *w*Pip groups. (B) CidA and CidB interact head-to-tail through 3 distinct interfaces I, II, and III, highlighted here in green, yellow, and red, respectively. The *cidA-III-β* and *cidB-IV-2* regions, which are key to the (in)compatibility between Slab and Ist isofemale lines, are both exposed in the interaction interface I. (C) Network phylogenies of the low-variability regions using the Neighbor-Net method on variants from this study and from [21,41,43]. Each edge (or set of parallel edges) corresponds to a split in the dataset with a length equal to the weight of the split. Similar regions are grouped within clusters. Within these clusters: (i) *cidB* regions are identical and are thus named the same way; (ii) in each given *cidA* cluster, sequences differ by at most 3 SNPs (located outside of any interface region). All variants within a cluster are named by the same Greek letter (alpha, beta. . .), with a number between brackets to reflect the existence of distinct variants (e.g., *cidA-I-α(1)* or *cidA-I-α(2)*). Recombinant variants and regions of interest in the study are highlighted in green. (D) Recombinations in key variants lead to original sequences at CidA-CidB interaction interfaces. Alignments of the protein sequences of the interaction interface I of CidA (up) and CidB (down) show that focal variants (in green) exhibit an original amino acid sequence. The sequence represented is the sequence between the first and last amino acid belonging to interface I (Wang and Hochstrasser, pers. com.). Accession numbers for sequences used in Fig 2D are found in S3 (previously sequenced variants) and S4 (newly sequenced variants) Tables. SNP, single-nucleotide polymorphism; *w*Pip, *Wolbachia* infecting *Culex pipiens*.

successfully analyzed (S1 Data). The PCR assay confirmed that 3 distinct *Wolbachia cidA* repertoires coexisted in the Slab isofemale line: (i) individuals hosting *Wolbachia* with both the upstream *β(2)* and downstream *16* regions, named (β+,16+); (ii) individuals hosting *Wolbachia* with only the upstream region *β(2)* but not the downstream *16* region, named (β+,16−); and (iii) individuals hosting *Wolbachia* genomes with neither of these regions, named (β−,16−) (S1 Data). Hereinafter, we refer to a mosquito by its *Wolbachia* infections' *cidA* repertoires

((β+,16+), (β+,16-), or (β-,16-)). Of the 89 females analyzed, 58 were (β+,16+), 8 (β+,16-), and 23 (β-,16-) (S1 Data).

Egg rafts produced by (β-,16-) females had a null HR, with a single exception, in which only 1 larva hatched, resulting in an HR of 1.8%. Fully incompatible crosses were only observed with (β-,16-) mothers, as all (β+,16+) and (β+,16-) mothers were compatible with Ist males (but with variability in HR) (Fig 1).

## *cidA* repertoire variations do not explain HR variability in compatible crosses

Examining the compatible crosses involving (β+,16+) or (β+,16-) Slab females and Ist males, we found strong variations in HRs ranging from 3.2% to 96.4% (Fig 1 and S1 Data) similar to those recorded in 2017 for which the HR ranged from 8% to 84% [41] (S1 Data). While infection levels (quantified by quantitative polymerase chain reaction (qPCR)) in full female DNA extracts were variable, with 0.11 to 10.1 *Wolbachia* genomes per host genome and an average of $1.0 \pm 0.2$ *Wolbachia* genome per host genome (mean ± standard error), this variability did not correlate with HR variations (S2A Fig, Spearman correlation test, $\rho = -0.09$, $p = 0.49$, S1 Data). Similarly, HR variations were not correlated with the *cidA* copy numbers (total *cidA*, *cidA-III-β*, or *cidA-III-16* copy numbers) per *w*Pip-Slab genome investigated in Slab females (Spearman correlation test, $\rho = -0.13$, $p = 0.28$; $\rho = 0.01$, $p = 0.94$; $\rho = -0.17$, $p = 0.16$ for the total *cidA*, *cidA-III-β*, and *cidA-III-16*, respectively; S2B–S2D Fig and S1 Data).

For each of the 3 *cidA* repertoires found in the original Slab isofemale line, we established 2 isofemale sublines. HR variations could also be observed in intra-line crosses within these newly established isofemale lines, with HRs ranging from 29% to 100% (S3 Fig and S2 Data). While HR variance was overall similar in all intra-line crosses (Levene test, $p = 0.05$), HRs were significantly more variable in "compatible" crosses between Slab females and Ist males than in intra-line crosses (Levene test, $p < 0.0001$).

## Recombination in *cid* variants leads to original CidA-CidB interaction zones

Recent works revealed that there were 3 interaction interfaces between CidA and CidB proteins, named interfaces I to III (Fig 2B, [23,24]). The upstream region *cidA-III-β(2)*, present in both (β+,16+) and (β+,16-), seems key to enable Slab females to rescue Ist males. This region corresponds to the interface I of the CidA protein (Fig 2). Looking closer at the protein sequences at this interface, we found that CidA-III-β(2) has an original combination of amino acids resulting from a recombination of the CidA-α and CidA-γ regions (Fig 2D). Building a phylogenetic network of all the described *cidA* upstream regions [21,22,41,43] confirmed that β-like regions were rare (only found in the *w*Pip group IV to which the Ist line belongs and Slab to date) and that they came from a recombination of the *cidA-γ*-like and *cidA-α*-like regions, with these last 2 regions being frequent and found in all *w*Pip groups surveyed to date (Fig 2C). We also showed that the *cidA-δ* region was recombinant.

The CidB-IV-2 downstream region, present in *w*Pip-Ist, had previously been associated with incompatibility when crossed with a wide range of females [22]. This region interacts with *cidA-β* at the interface I (Fig 2B). When looking at wider phylogenetic scales based on all the previously published sequences, we found a clear recombinant pattern with 2 downstream regions shared among *w*Pip groups (CidB-1 and CidB-3) and a unique intermediate recombinant region, only found in *w*Pip group IV so far, CidB-IV-2 (Fig 2C).

## Evolution of *cid* repertoires

**Different *cidA* repertoires segregate into sublines.** Since a new CI pattern emerged between 2017 and 2021, a closer look at the transmission and variation of Slab individuals' repertoires was required to better understand the precise nature of this evolution. We sequenced the *cidA* and *cidB* repertoires of individuals in newly established isofemale sublines and confirmed that repertoires staid identical to those previously found for (β-,16-), (β+,16-), and (β+,16+) individuals. We confirmed their expected CI phenotype when crossed with Ist males (S4 Fig and S3 Data). We also verified that they were compatible with each other in both directions. We found that HRs were highly variable in those crosses too, with HRs ranging from 2% to 99%, but that there was no full incompatibility (S5 Fig and S4 Data).

To test whether changes in repertoires could occur at the scale of a few host generations, we genotyped *Wolbachia* from 20 to 45 individuals of each Slab subline after about 15 generations for both *cidA* markers and found that no change had occurred since the establishment of the isofemale sublines.

We also genotyped individuals from the initial 2005 Slab isofemale line (kept in liquid nitrogen since 2005), i.e., shortly after the establishment of the isofemale line in our insectaries. Out of 30 screened mosquitoes, there were 26 (β+,16-) and 4 (β+,16+). None of them was (β-,16-), which is in accordance with the hypothesis that this genotype, leading to a new CI pattern, appeared afterwards.

**Multiple copies of the same variant in the repertoires.** qPCR results revealed that the total number of *cidA* copies was always higher than the number of distinct variants found in a given repertoire, showing that some variants must be present in multiple identical copies (S6A Fig). Our previous work showed that reads coverage was informative of gene copy numbers when using Nanopore sequencing of amplicons [42]. By combining the qPCR data and relative read coverage information from Nanopore sequencing (e.g., variant with 2 copies is covered twice as much as variant with a single copy), we estimated the copy numbers of each variant in the different *cidA* repertoires of the Slab sublines. We found that (β+,16+) individuals, which have a qPCR total of 5 *cidA* copies, had 2 copies of *cidA-III-α(5)-25* and a single copy of each of the other 3 variants (*cidA-III-β(2)-16*, *cidA-III-β(2)-25*, and *cidA-III-γ(3)-12*). (β+,16-) individuals also had 5 *cidA* copies, of which there were 2 copies of *cidA-III-α(5)-25*, a single copy of *cidA-III-β(2)-25*, and 2 copies of *cidA-III-γ(3)-12*. Lastly, (β-,16-) individuals, which had a total of 3 *cidA* copies, had 2 copies of *cidA-III-α(5)-25* and a single copy of *cidA-III-γ(3)-12* (S7 Fig).

We confirmed the deduced copy numbers using qPCR specific to *cidA-III-β(2)* and *cidA-III-16*: quantifying these 2 markers in 89 single Slab females, we found that (β+,16+) repertoires had a single copy of *cidA-III-16*, while the other repertoires had none (S6B Fig). For *cidA-III-β*, we found that (β+,16+) individuals had 2 copies (corresponding to 1 copy of *cidA-III-β(2)-16* and 1 copy of *cidA-III-β(2)-25*), while (β+,16-) individuals had a single copy (unique copy of *cidA-III-β(2)-25*). Finally, (β-,16-) individuals had none of these copies (S6C Fig).

***cidA* and *cidB* variants are always located within tandems.** Previous investigations describing the *cid* repertoire of a *w*Pip in a given *Culex* line were performed by acquiring the sequences of *cidA* and *cidB* variants separately [21,22,41–43]. Here, we also amplified and Nanopore sequenced the *cidAB* region, encompassing both *cidA* and *cidB* variable regions in a single amplicon (S8 Fig). We could thus reconstruct the *cidA-cidB* tandems associated in the genome, resulting in complete repertoires (S7 Fig). To determine whether all *cid* variants were associated in tandems or whether they could stand alone in the genome, we compared their relative coverage when amplified and sequenced alone versus when amplified and sequenced

in *cidA*-*cidB* tandem acquisitions. As these relative coverages were similar when *cidAB* repertoires were sequenced, and when *cidA* or *cidB* repertoires were sequenced alone (e.g., for a (β-,16-) individual: 10,800 reads for *cidA-III-γ(3)-12* and 32,000 for *cidA-III-α(5)-25* in the *cidAB* repertoires versus 3,000 and 10,000 in the *cidA* alone repertoire), we considered that all *cidA* and *cidB* variants are always located in tandems. Thus, (β+,16+) and (β+,16-) sublines have 5 *cidAB* tandems, while (β-,16-) lines have 3 *cidAB* tandems (S7 Fig). Individuals of all sublines have *Wolbachia* with 2 copies of the *cidA-III-α(5)-25_cidB-III-ae3* tandem and 1 copy of the *cidA-III-γ(3)-12_cidB-III-ag1* tandem. In addition, (β+,16+) and (β+,16-) individuals have 1 *cidA-III-β(2)-25_cidB-III-ae3* tandem, although they differ in terms of their fifth tandem: *Wolbachia* in (β+,16+) individuals have a specific *cidA-III-β(2)-16_cidB-III-ae3* tandem, while those in (β+,16-) individuals have a *cidA-III-γ(3)-12_cidB-III-ae3* tandem instead (S7 Fig). Our results also show that the *cidB* variant *"cidB-III-ae3"* is associated in tandem with 4 different *cidA* variants (S7 Fig).

**No sign of stable coinfection.** We found that repertoires of the different sublines had similar variants and the same tandem architectures. We have previously described a very high variability of repertoires among the distinct *w*Pip strains studied [21,22,41]. The fact that Slab sublines here only differ by the presence/absence of 1 or 2 variants, and share an identical architecture strongly supports the evolution hypothesis (rather than contamination). We used qPCR on Slab females to look at gene copy number distribution. We found that total *cidA* copy numbers were variable, ranging from 2.2 to 6.2 copies per *w*Pip genome (quantifications normalized on the single copy *Wolbachia* gene *wsp*; S6A Fig). However, the distribution was bimodal with 2 peaks around 3 and 5 *cidA* copies (S6D Fig). The *cidA* copy number was homogeneous for females infected with a given *cidA* repertoire, with around 5 *cidA* copies for all (β+,16+) and (β+,16-) individuals and around 3 *cidA* copies for all (β-,16-) individuals (S6D Fig). We also quantified the number of *cidB* copies in 8 to 10 females of each repertoire and found that they were similar to *cidA*: around 5 copies in both (β+,16+) and (β+,16-) individuals and around 3 *cidB* copies in (β-,16-) individuals (S9 Fig). These results show that while Slab *cidB* repertoires do not differ between lines in terms of the identity of variants, they differ in terms of the *cidB* copy numbers. The centered data distributions in S6D–S6F Fig suggest that most individuals (if not all) from all the (β+,16+), (β+,16-), and (β-,16-) sublines are not coinfected by *Wolbachia* with different *cid* repertoires.

To check whether the variability in qPCR results could be explained by the intrinsic variability of the qPCR method, we quantified twice independently the *cidA* copy numbers in a (β+,16+) mother and 20 of her offspring (extracted at the larval stage). We found that the *cidA* copy numbers are centered around 5 in both measurements, but that the first measurement and the second one are not correlated (S10 Fig, Pearson's correlation test, $p = 0.14$), demonstrating the qPCR-linked variability and supporting the absence of coinfection.

## Discussion

The evolution of *Wolbachia* genomes has already been documented at large phylogenetic scales, with multiple horizontal transfers and gene losses (e.g., [33,34,39,44–47]), especially in the *cif* gene family in which numerous horizontal transfers have been described [25,26,30]. Here, we observed rapid changes in CI patterns with the recent impossibility for some Slab females, within the same isofemale line, to rescue Ist males. We show that this modification in rescue phenotype is directly linked to the loss of a *cidA* variant containing the specific *cidA-III-β* region. The combination of long-term mosquito rearing, regular crosses and analyses of underlying genes used here brings new support for the TA model of CI in *w*Pip in a natural context.

## Rapid evolution of *cid* repertoires through gene losses

In 2021, there were approximately one quarter of females which had 3 *cidA-cidB* tandems. These three-copy females were absent in the original line in 2005, showing that 2 *cid* tandems had been lost since the line's establishment. *cid* copy number distributions are bimodal, with 2 peaks at 3 and 5 copies and very rare with intermediate repertoires of 4 copies. These rare intermediates are likely due to the qPCR variability. This could mean either that (i) four-copy repertoires were a transient state that disappeared quickly due to lower fitness or drift or that (ii) four-copy repertoires did not exist, meaning that 2 tandems were lost in a simultaneous genomic structural variation.

While the loss of 2 tandems affects *cidA* and *cidB* numbers similarly, we observed a decrease in the number of distinct *cidA* variants without a change in the *cidB* repertoires: all (β+,16+), (β+,16-), and (β-,16-) repertoires described here share the same *cidB* variants (*cidB-III-ag1* and *cidB-III-ae3*), while they have 4, 3, and 2 distinct *cidA* variants, respectively (Fig 1). Literature on CI evolution suggests that *mod* factors or toxins (*cifB* variants) could degenerate first, as their evolution is neutral for maternally transmitted *Wolbachia* [48,49]. Only once *cifB* variants have lost their function is it possible for the loss of *cifA* (*resc* factors or antidotes) [50]. In such a conceptual framework, losing *cifA* without losing *cifB* would lead to self-incompatibility, which is an immediate evolutionary dead-end. Independent from CI, the presence of *cidB* without an associated *cidA* might be toxic for the host, as shown recently in insect cells [4]. These predictions have been corroborated by a large-scale phylogenetic study of *cif* genes, indicating that *cifB* genes were often disrupted, while *cifA* were not [26]. This apparent discrepancy with the predictions may be explained by the fact that *w*Pip genomes contain several *cid* genes. Unlike conceptual framework where the loss of *cifA* was thought to lead to a loss of the self-rescuing ability, *w*Pip has several *cidA* variants, which makes it possible to lose 1 variant while still maintaining compatibility with the surrounding mosquitoes.

Laboratory conditions could facilitate this loss of antidotes. Indeed, in natural populations, individuals with multiple *cid* repertoires coexist in the same population (as shown by sequencing and revealed by numerous crosses [21,22,43,51]), meaning that crosses can involve a wide range of toxins. A greater diversity in the *cidA* repertoire enables the rescue of a wider range of toxins, which is thus under positive selection. While *cidA* diversity is advantageous in natural populations, in an isofemale line cage, crosses are involving a restricted and fixed set of toxins. Loss of *cidA* variants, which are not strictly required for the rescue of the *cidB* in a given cage, is not counter-selected. Furthermore, as numerous gene copies can be costly (e.g., [52]), the loss of *cidA-cidB* tandems could confer a higher fitness to *Wolbachia* with lower numbers of copies and thus contribute to the fixation of new *Wolbachia* genomes in mosquitoes, which could explain the rapid *Wolbachia* turnover. It is conceivable that the three-copy repertoire of (β-,16-) *Wolbachia*, which contains 2 distinct variants of *cidA* and 2 distinct variants of *cidB*, is the minimum viable repertoire with only pairs of matching toxins and antidotes.

## No lasting coinfection involved in changes in CI patterns in *w*Pip

One plausible explanation for the observed shift in CI could be that Slab is coinfected by 2 strains of *Wolbachia* with different *cid* repertoires, which are usually co-transmitted, and that some individuals recently lost one of these strains, as observed in *Aedes albopictus* [53,54]. Using common MLST markers, Baldo and colleagues initially found no variability in the *w*Pip strains [55], showing that if coinfection occurs, it should be between highly similar strains. The development of a *w*Pip-specific MLST led to the uncovering of *w*Pip polymorphism and the description of 5 monophyletic groups within *w*Pip [56]. However, while these markers highlighted *w*Pip polymorphism, no sign of coinfection was found, as every mosquito clearly

hosted one of the 5 groups [56]. In this study focusing on *cid* genes, which are highly variable even within a *w*Pip group, we likewise found no sign of coinfection. We quantified several *cid* markers and analyzed the distribution of their values. In the case of coinfections with 2 *Wolbachia* strains with 3 and 5 *cid* copies, respectively, quantification outcomes would have varied depending on the proportion of each *Wolbachia* in each mosquito, leading to a unimodal distribution centered on an intermediate value. For the 3 distinct *cid* markers analyzed in 89 females, the quantifications displayed a bimodal distribution (S6D–S6F Fig), with a few intermediate values probably due to technical variations in the qPCR measures (S10 Fig). While coinfection must occur at some point (all *Wolbachia* within one individual cannot be wiped out and instantly replaced by another *Wolbachia*), it had to be brief with no lasting coinfection in a single host individual. This could be explained by 2 alternative hypotheses: (i) an exclusion between the 2 distinct genotypes (3 and 5 copies) for any non-CI-related reason; or (ii) a small effective population of *Wolbachia* (number of *Wolbachia* transmitted from mother to offspring) in *w*Pip.

## Variability in hatching rates in compatible crosses not explained by *cid* repertoires of females

Studying the cross between Slab females (*w*Pip-III) and Ist males (*w*Pip-IV), we found intermediate HRs in "compatible" crosses. While on/off compatibility (hatching versus no hatching at all) was clearly explained by the presence/absence of specific *cidA* variants, these intermediate HRs did not correlate with any of the *cidA* markers nor with the *Wolbachia* infection rate. Several factors could contribute to such variable HRs in "compatible" crosses: (i) some *Wolbachia*-independent effects, with loss of fertility in these strains due to inbreeding following a prolonged lab rearing (as previously noted by, e.g., [57]); and (ii) some *Wolbachia*-linked effect, with partial incompatibilities, as we recently described, for the first time in *C. pipiens* [41]. The first hypothesis is supported by previous findings, showing variable HRs even in crosses between *Wolbachia*-devoid mosquitoes [41]. This decrease in fertility could also be combined with a *Wolbachia*-linked effect. Indeed, we previously showed that male mosquitoes infected with a *Wolbachia* from *w*Pip group IV (such as Ist males) induced more partial incompatibilities [41]. Furthermore, it could be envisaged that there are *cidB* (toxin) copy number variations in males, which we could not detect here since our protocol isolated the females to lay their eggs and did not track the males. We also found that intra-line crosses exhibited variable HRs, but that this variability was significantly higher in Slab x Ist crosses, strongly suggesting *Wolbachia*-linked effects.

## Recombination as the key to the emergence of new CI patterns

The presence/absence of a specific *cidB* region, named *cidB-IV-2*, was previously correlated with the ability or not of males to sterilize a wide range of females [21,22]. This specific *cidB-IV-2* region is present in the genome of the *Wolbachia* infecting the Ist males studied here. We showed that this specific region is issued from a recombination between the 2 main groups of *cidB* downstream regions (1 and 3), which are shared by all sequenced *w*Pip groups (Fig 2).

Furthermore, it was previously shown that the only females able to rescue the modifications induced by Ist males were those infected with *w*Pip-IV (as Ist) as well as Slab females (*w*Pip-III, studied here) [21,22]. We explained the rescuing ability of Slab females by the presence of a specific *cidA* upstream region known as *cidA-β*: all females with this region can rescue Ist males, while females that have lost it cannot. This variant was also generated by a recombination between 2 main groups of upstream regions shared by all *w*Pip groups. Here, there were 2 types

of recombinants, the *cidA-β* and *cidA-δ* regions (Fig 2). All these recombinant regions were previously found in field populations, confirming they are not lab oddities [21,22].

When looking at the repertoires found in females infected with *w*Pip-IV, which can all rescue Ist males, we found that they all had 1 recombinant region in their repertoire (either *cidA-β* or *cidA-IV-δ*) [21,22]. In other words, all females from field populations or laboratory lines able to rescue a toxin containing the recombinant region *cidB-IV-2* have a recombinant *cidA-β* or *cidA-δ* antidote in their repertoire. Since the *cidA* upstream region (e.g., *β* or *δ*) interacts with the *cidB* downstream region (e.g., *cidB-IV-2*) in the interaction interface I [24], this points to recombinant-for-recombinant binding (Fig 2). New CI phenotypes could thus emerge through the recombination of existing variants. As it demonstrates a key role for binding in compatibility, this finding strongly supports the toxin–antidote hypothesis, for the first time in a natural system.

In conclusion, we found that CI evolved in less than 4 years in a *Culex* isofemale line. This evolution is linked to the loss of some *cidA* (antidote) variants in their *cid* repertoire. Our results lead to propose how new incompatibilities could emerge as a two-step process in the wild: first, local changes take place in the CI repertoires while maintaining compatibility with the surrounding mosquitoes, and then migration and secondary contact occur with the incompatible lines. Here, we showed that all repertoires present in a cage were mutually compatible: changes in the repertoires modified the CI pattern only when crossed with an external line. We demonstrated that new compatibility types can emerge rapidly, as a result of the rapid evolution of *Wolbachia* genomes. A common point in the few studies addressing *Wolbachia* microevolution is that both Octomom in *w*Mel [37–39] and *cid* genes in *w*Pip are amplified genomic regions, suggesting the strong role played by gene amplification in the speed of *Wolbachia* genome evolution. This rapid evolution should therefore be considered when releasing *Wolbachia*-infected mosquitoes, inducing CI, for vector control purposes.

## Materials and methods

All DNA extractions in this study were performed following the cetrimonium bromide (CTAB) protocol [58].

### Isofemale lines

All the mosquito lines used were isofemale lines, i.e., lines created by rearing the offspring resulting from a single egg raft and thus from a single female. We used the following lines: Slab, infected with *Wolbachia* from the phylogenetic group *w*Pip-III, and Istanbul (Ist), infected with *Wolbachia* from the phylogenetic group *w*Pip-IV. Both lines were established in our laboratory in 2005.

We also used 6 isofemale (sub)lines, which we established for this study in 2021: 2 (β+,16+) lines bearing both the *cidA-III-β* and *cidA-III-16* regions; 2 (β+,16-) lines bearing only the *cidA-III-β* marker; and 2 (β-,16-) lines with neither of these markers. In short, to establish the isofemale line, a cage containing the Slab isofemale line was blood fed. After 5 days, females were isolated in glasses for egg laying. After egg laying, DNA was extracted, and females were genotyped for *cidA-III-β* and *cidA-III-16* (S2 Table). For each genotype, the 2 most numerous progenies were kept in order to establish 2 separate isofemale lines.

All isofemale lines were reared in 65 dm$^3$ screened cages in a single room maintained at 26˚C under a 12-h light/12-h dark cycle. Larvae were fed with a mixture of shrimp powder and rabbit pellets, and adults were fed on a honey solution. Females were fed with turkey blood using a Hemotek membrane feeding system (Discovery Workshops, United Kingdom) to enable them to lay eggs.

## Crosses

For each cross performed, around 100 virgin females were put in a cage with around 50 virgin males. After 5 days in cages, females were fed a blood meal, and 5 days later, individual females were isolated for egg laying. This allows us to precisely link a female with its egg raft. Egg rafts were deposited into 24-well plates. The eggs were photographed using a Leica camera and counted using ImageJ. Two days after egg laying, larvae issued from each egg raft were counted manually. HR was calculated as the proportion of eggs that hatched into larvae. Egg rafts with a null HR were mounted on a slide and examined with a binocular magnifier to ensure that the eggs were fertilized and thus confirm that null HR resulted from CI rather than non-fertilization.

## Repertoire acquisition using Nanopore sequencing of PCR products

We acquired the *cidA* and *cidB* repertoires of 8 individuals from the Slab isofemale line, 4 individuals from the Ist isofemale line, and 2 individuals from each of the newly established isofemale lines following the pipeline described in [42]. We also acquired *cidAB* combinations for all newly established isofemale lines. DNA was extracted from the individuals. In short, the target sequence was amplified using the adequate primer set (S2 Table and S8 Fig). PCR products were then sequenced by Nanopore sequencing with the MGX Platform (Montpellier) using a MinION (Oxford Nanopore Technologies). This gave an average coverage of 30,000 to 50,000 reads for each gene. Repertoires were then acquired following [42]. Briefly, reads were mapped on a reference base containing all the variants previously identified, along with variants created in silico. Single-nucleotide polymorphism (SNP) calling was then performed, along with haplotype phasing if necessary. For *cidAB* combinations, the pipeline was adapted: *cidA* and *cidB* repertoires were first obtained, and then the combinations between all *cidA* and *cidB* were artificially created and used as a reference base to run the pipeline.

## *Wolbachia* genotyping

**Standard PCR.** To genotype individuals, DNA was extracted. The presence/absence of both *cidA-III-16* and *cidA-III-β* was determined using specific PCRs (S2 Table), using the GoTaq G2 Flexi Polymerase (Promega). For individuals negative for both markers, a PK1 PCR (S2 Table) was used to control DNA quality. This was performed on either adult females or fourth instar larvae.

**Quantitative PCR.** Real-time quantitative PCR was run using the LightCycler 480 system (Roche) and the SensiFAST SYBR No-ROX Kit (Meridian Biosciences). All DNA samples were analyzed in triplicates for each quantification. A total of 89 adult females were used for qPCR assays.

The total number of *cidA* copies, along with the number of *cidA-III-β* and *cidA-III-16* region copies, was quantified using relative quantitative PCR, corrected by *wsp*, a single-copy *Wolbachia* gene (following [59], primers in S2 Table). Since *wsp* is present in a single copy per haploid genome, the ratio of the target and *wsp* signal can be used to estimate the number of target copies per *Wolbachia* genome, correcting for *Wolbachia* infection level and DNA quality. For *cidA-III-β* qPCR, quantifications were normalized on 3 distinct individuals with a single *cidA-III-β* copy.

The infection level (i.e., the number of *Wolbachia* bacteria per female) was assessed with qPCR using a *wsp* PCR normalized on the *Culex*-specific single-copy *ace-2* locus [60]. The ratio between *wsp* and *ace-2* signals can be used to estimate the relative number of *Wolbachia* genomes per *Culex* genome.

For *cidA-III-16/wsp* and *cidA/wsp*, the following protocol was used: 3 min at 95˚C, 45 cycles of 95˚C for 10 s, 60˚C for 20 s, and 72˚C for 20 s. For *cidA-III-β/wsp* and *ace2/wsp*, the protocol was as follows: 3 min at 95˚C, 45 cycles of 95˚C for 3 s, 62˚C for 10 s, and 72˚C for 15 s. Primer concentrations were 0.6 μm for *cidA-III-16* and *cidA*, 0.4 μm for *wsp* and *ace2*, and 1 μm for *cidA-III-β*.

### Variant copy number analysis

Variant copy numbers were obtained using both the *cidA* qPCR data and the relative coverage of Nanopore sequencing. Knowing the total number of *cidA* copies, we deduced each variant copy number. For example, in the case of 3 *cidA* copies, if out of the 2 variants present, the coverage of the first was twice the coverage of the second, it could be concluded that there were 2 copies of the first variant and 1 copy of the second.

### Grouping analysis of upstream/downstream regions of variants

Phylogenetic analysis of variants was performed using all the variants already published [21,22,41,43] and present in this study. Network phylogenies of the 2 regions of interest (*cidA* upstream variable region and *cidB* downstream region, which respectively corresponded to nucleotides 1–800 and 1201-end) were made using Splitstree [61].

### Nomenclature of *cid* variants

The nomenclature established by [21] was updated, since it could result in identical sequences being given different names in different *w*Pip groups. The fundamentals of the previous nomenclature were retained: names of variants were composed of the name of their upstream region (in Greek letters for *cidA* and in Latin letters for *cidB*) and that of their downstream region (digits). A fixed limit was established between the upstream and downstream regions for both genes (downstream regions starting at nucleotides 801 and 1201 for *cidA* and *cidB*, respectively). A phylogenetic network revealed 4 clusters of *cidA* upstream regions (Fig 2C), regions within a cluster differing by at most 3 SNPs. We renamed regions so that all those belonging to the same cluster were named using the same Greek letter (e.g., all named α). When SNPs were present, to distinguish the sequences, we introduced a number between brackets such as α(1). The same was done for *cidB* downstream regions. Similarly, *cidA* downstream regions and *cidB* upstream regions were renamed so that distinct regions had different names, and identical regions had identical names, regardless of the *w*Pip group in which they were found.

For all variant names, the group in which the variant occurs is mentioned. For example, cidA-III-α(5)-25 is found in a *w*Pip-III.

The correspondence between the previously used names and the names based on this new nomenclature is provided in S3 Table.

### Statistical analyses and graphical representation

All statistical analyses were run using R version 4.2.0 [62]. Correlation analyses were performed using a Pearson correlation test. Homoscedasticity was tested using Levene tests. All boxplots and histograms were done using ggplot2 [63], and sequence representation was conducted using the function msaPrettyPrint from the msa package [64]. HRs analysis were done using a generalized linear model (GLM), with a binomial distribution. The significance of the different terms was testing using likelihood ratio tests (LRT).

## Supporting information

**S1 Fig. Hatching rates distribution for crosses between Slab females and Istanbul males when crossed at the lab in 2017 (left) and in 2021 (right).** A notable difference is the existence of crosses with null hatching rate in 2021. Data supporting this figure are found in [41] (for 2017 Data), and in S2 Data (2021 Data).
(TIF)

**S2 Fig. Variations in hatching rates in compatible crosses between Slab females and Istanbul males.** Hatching rates in relation to variations in: (A) *Wolbachia* infection level and numbers of copies of (B) total *cidA*, (C) *cidA-III-β(2)*, and (D) *cidA-III-16*. None of these variations are significantly correlated with hatching rates (Spearman correlation coefficient, $P = 0.49$, 0.28, 0.94, and 0.16 for (A–D), respectively). Data supporting this figure are found in S1 Data.
(TIF)

**S3 Fig. Intra-line crosses in newly established isofemale lines.** For each line, 20 to 30 egg rafts were collected and the hatching rate was measured. Data supporting this figure are found in S2 Data.
(TIF)

**S4 Fig. Crosses between females from each of the 6 newly established isofemale sublines (2 (β+,16+), 2 (β+,16-), and 2 (β-,16-)) and Istanbul males.** Data supporting this figure are found in S3 Data.
(TIF)

**S5 Fig. Reciprocal crosses between (β+,16+) and (β-,16-) isofemale sublines.** Data supporting this figure are found in S4 Data.
(TIF)

**S6 Fig. Variations of *cidA* copy numbers in *Wolbachia* genomes of Slab females.** (A–C) show the numbers of copies of total *cidA*, *cidA-III-16*, and *cidA-III-β(2)* regions, respectively, in females with repertoires (β-,16-), (β+,16-), and (β+,16+). (D–F) show the distribution of *cidA*, *cidA-III-16*, and *cidA-III-β(2)* copy numbers. All histograms show either bi- or trimodal distributions. All copy numbers were quantified using quantitative PCR relative to the single copy *Wolbachia* gene *wsp*, on 23 (β-,16-), 8 (β+,16-), and 58 (β+,16+) adult females. Data supporting this figure are found in S1 Data.
(TIF)

**S7 Fig. Complete *cid* repertoires for the 3 Slab distinct sublines.** The *cidA* gene is represented in light purple and *cidB* in dark purple. Upstream and downstream variable regions are shown in color. Regions in green are the key studied regions: *cidA-III-β(2)* and *cidA-III-16*. The copy number of each tandem was deduced by combining the relative coverages in nanopore sequencing data with the qPCR data.
(TIF)

**S8 Fig. PCR primers used to analyze sequence variation in *cid* genes.** PCRs amplifying the full *cidA* gene (burgundy arrows), variable regions of the *cidB* gene (green arrows), and the *cidAB* tandem (purple arrows) were used for Nanopore sequencing of repertoires. PCRs specific to *cidA-III-β(2)* (yellow) and *cidA-III-16* (orange) were used both in standard PCR and qPCR to determine the presence/absence of these regions.
(TIF)

**S9 Fig. Variations of *cidB* copy numbers in *Wolbachia* genomes of Slab females.** *cidB* copy numbers are similar to *cidA* copy numbers. (β-,16-) *w*Pip has around 3 *cidB* copies, whereas (β+,16-) and (β+,16+) have around 5 *cidB* copies. All copy numbers were quantified using quantitative PCR relative to the single copy *Wolbachia* gene *wsp*. Data supporting this figure are found in S1 Data.
(TIF)

**S10 Fig. Quantitative PCR (qPCR)-induced variation in copy number measurements.** *cidA* copy numbers from the same individuals were measured twice (measure 1, measure 2) in 2 distinct qPCR runs. While all measurements are around 5 copies, there is no significant correlation between the first and the second measurement. Data supporting this figure are found in S5 Data.
(TIF)

**S1 Table. *cid* repertoires of the Istanbul isofemale line.** Nanopore sequencing of *cid* PCR products gave identical repertoires for 4 distinct Istanbul individuals.
(DOCX)

**S2 Table. PCR primers and programs used in this study.** qPCR programs are described in the main text.
(DOCX)

**S3 Table. Correspondence between variant names in the previous (Bonneau and colleagues) and current nomenclature.**
(XLSX)

**S4 Table. GenBank accession numbers of new *cidA* and *cidB* variants used in this study.**
(XLSX)

**S1 Data. Crosses between Slab females and Istanbul males.** For each female, we determined the hatching rate (HR) (i.e., the proportion of eggs that hatched into larvae). The specific *cidA-III-β(2)* upstream and *cidA-III-16* downstream regions were quantified in each female and normalized over the single copy *Wolbachia* gene *wsp*. The total *cidA* and *cidB* copy numbers were also quantified and normalized over *wsp*. Infection rates were assessed by quantifying the single-copy gene *wsp* and normalizing it on the *Cx. pipiens* single-copy gene *ace2* (Berticat and colleagues).
(XLSX)

**S2 Data. Intra-line crosses for each of the 6 newly established isofemale lines.** For each line, 20 to 30 egg rafts were collected and the hatching rate was measured.
(XLSX)

**S3 Data. Crosses between females from each of the 6 newly established isofemale sublines (2 (β+,16+), 2 (β+,16-), and 2 (β-,16-)) and Istanbul males.** The hatching rates are shown for each female. Eggs were only counted in compatible crosses involving (β+,16+) and (β+,16-) females. All females were screened for the presence/absence of 2 regions of the polymorphic gene *cidA* (*cidA-III-β(2)* and *cidA-III-16*) to confirm that they had the same repertoire as the female from which the isofemale subline was founded. NF = non-fertilized.
(XLSX)

**S4 Data. Reciprocal crosses between (β+,16+) and (β-,16-) isofemale sublines.** Number of eggs and larvae and resulting hatching rates (HR). NF = non-fertilized.
(XLSX)

**S5 Data. Quantitative PCR (qPCR)-induced variation in copy number measurements.** *cidA* copy numbers from the same individuals were measured twice (measure 1, measure 2) in 2 distinct qPCR runs. Measurements and measurement errors are given.
(XLSX)

## Acknowledgments

We thank Michael Turelli, Nicole Pasteur for their thorough reading and helpful comments on the manuscript. We thank Pierrick Labbé for statistical advice, and Wei Wang and Mark Hochstrasser for their input on the interface residuals. Sequencing data were generated at the Montpellier GénomiX platform.

## Author Contributions

**Conceptualization:** Alice Namias, Mathieu Sicard, Mylène Weill.

**Funding acquisition:** Mylène Weill.

**Investigation:** Alice Namias, Annais Ngaku, Patrick Makoundou, Sandra Unal.

**Methodology:** Patrick Makoundou.

**Supervision:** Mathieu Sicard, Mylène Weill.

**Writing – original draft:** Alice Namias, Mathieu Sicard, Mylène Weill.

**Writing – review & editing:** Alice Namias, Mathieu Sicard, Mylène Weill.

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
