## [Editor Report · Decision Letter 0]

14 Jul 2023

Dear Dr Namias, 

Thank you for submitting your manuscript entitled "Intra-lineage microevolution of Wolbachia leads to the emergence of new cytoplasmic incompatibility patterns" for consideration as a Research Article by PLOS Biology.

Your manuscript has now been evaluated by the PLOS Biology editorial staff, as well as by an academic editor with relevant expertise, and I'm writing to let you know that we would like to send your submission out for external peer review.

IMPORTANT: Although you submitted your paper as a regular Research Article, we think that it would be better considered as a Short Report. Please change your article type to "Short Report" when you upload your additional metadata (see next paragraph).

Once your full submission is complete, your paper will undergo a series of checks in preparation for peer review. After your manuscript has passed the checks it will be sent out for review. To provide the metadata for your submission, please Login to Editorial Manager (https://www.editorialmanager.com/pbiology) within two working days, i.e. by Jul 18 2023 11:59PM.

Kind regards,

Roli Roberts

Roland Roberts, PhD

Senior Editor

PLOS Biology

rroberts@plos.org

---

## [Decision Letter · Decision Letter 1]

31 Aug 2023

Dear Dr Namias,

Thank you for your patience while your manuscript "Intra-lineage microevolution of Wolbachia leads to the emergence of new cytoplasmic incompatibility patterns" was peer-reviewed at PLOS Biology. It has now been evaluated by the PLOS Biology editors, an Academic Editor with relevant expertise, and by three independent reviewers. 

You’ll see that reviewer #1 is very positive about the paper, but has a long list of presentational improvements and requests for clarification; some of these may involve new data. Reviewer #2 says it’s interesting, but currently better suited to a specialised audience. His/her main concern is expressed in “point 1,” which is a list of missing aspects; while these would clearly improve the study, we think that supplying all of them may not be reasonable for a Short Report (see Academic Editor's comments below). Her/his “point 2” complains about the frustratingly arcane nomenclature (I totally agree!) and makes some presentational suggestions. Reviewer #3 does not comment on the advance, but has several requests for methodological clarity, wants some formal stats, finds the abbreviations confusing, and has a number of presentational suggestions.

IMPORTANT: Given the divergent opinions on this paper, I discussed the reviews with the Academic Editor who kindly provided some additional guidance which you may find helpful when deciding how to revise the paper, and which you should address directly:

"An issue pointed out by the reviewers, and that I agree with, is that the paper is difficult to read and would need to be re-written to a certain extent.

"Another common point is that some of the data that is discussed should be presented, as for example, compatibility data between the lines in the beginning and when they noticed that there was a change. Also, self-compatibility data is missing, and would help to understand better the results. Self-compatibility data may have to be generated and may be difficult to obtain in a short time.

"I think the fitness of the new variant is not an essential aspect of this paper as a short report. But reviewer #2's point 1 request may be related to how the paper is written and presented. The abstract ends on speculation on evolutionary and ecological dynamics of CI and trying to project these findings in a lab population to natural populations dynamics. The fact that these genes can be easily lost, is very interesting with many possible consequences. But maybe the main ecological and evolutionary conclusions should be presented more as speculation and less as conclusion. I think the authors don’t discuss for example why are these genes maintained in Wolbachia in a natural population and how different is a natural population and and an isofemale line with one founding Wolbachia.

"For me the strength of the paper is more on having a specific phenotype change in CI associated with a specific change in genotype in CI genes. This is actually a very strong case, supported on Wolbachia genetics, for how CI works. All the previous great work is mostly supported by transgenes assays. More, I would argue, which the authors don’t, that the findings are a great support for the toxin-antitoxin model. A very specific recombination between two CidB is only rescued by a recombination between two CidA that seems to restores sequence compatibility.

"Overall, I agree with the major revision and see how the authors respond to the comments. I would stress:

1) the re-writing to make the paper easier to understand

2) presenting more of the data that is discussed

3) more clarity in what is shown and possible ecological and evolutionary dynamics consequences."

In light of the reviews, which you will find at the end of this email, we would like to invite you to revise the work to thoroughly address the reviewers' reports.

Given the extent of revision needed, we cannot make a decision about publication until we have seen the revised manuscript and your response to the reviewers' comments. Your revised manuscript is likely to be sent for further evaluation by all or a subset of the reviewers.

**IMPORTANT - SUBMITTING YOUR REVISION**

*Re-submission Checklist*

*Published Peer Review*

*PLOS Data Policy*

*Blot and Gel Data Policy*

Sincerely,

Roli Roberts

Roland Roberts, PhD

Senior Editor

PLOS Biology

rroberts@plos.org

REVIEWERS' COMMENTS:

Reviewer #1:

Wolbachia bacteria are widespread in insect populations and can influence their reproduction through cytoplasmic incompatibility. Culex pipiens mosquitoes have diverse Wolbachia infections which coincides with a variety of crossing types, where different Wolbachia variants induce cytoplasmic incompatibility with some variants but not others. While the genes responsible for cytoplasmic incompatibility have been identified, the basis of this crossing type diversity is not fully understood and it is unclear how it evolved. Here, the authors describe the emergence of a Wolbachia variant with a new cytoplasmic incompatibility type over a very short period (<4 years) in a laboratory colony. This came from an observation that a cross between two previously compatible laboratory lines (Slab and Ist) were now partially incompatible, with around 25% of females in this cross producing zero viable offspring. They then established isofemale lines and performed sequencing to show that this was due to the loss of a variant of the CidA gene, which in the female is responsible for restoring compatibility with Wolbachia-infected males that have a different crossing type. Their experiments clearly demonstrate that these females have lost their ability to restore compatibility rather than the males evolving a new crossing type. This study appears to be the first case where the evolution of crossing patterns has been observed in real time and linked directly to a change in genotype. The paper is a significant advance in our understanding of Wolbachia evolution and has implications for mosquito control programs which rely on the release of Wolbachia-infected mosquitoes to induce cytoplasmic incompatibility to either suppress or replace local mosquito populations. While the paper is well written and the experiments are robust, I have several suggestions below:

Figure 1 shows outcomes from a cross between Slab females and Istanbul males then separates hatch rates by genotype of the females. The hatch rates for the compatible crosses are highly variable - is it possible that these lines are partially incompatible? No data for within-strain crosses (e.g. Slab x Slab) are shown but the hatch rates for these would likely be much higher and more consistent. This is also the case for the data in tables S3 and S4 where within line crosses were not completed or shown. While this doesn't affect any of their conclusions it's a shame that these were left out as it would help to confirm whether these variable hatch rates were due to cytoplasmic incompatibility, fitness costs or some other reason.

For future work, have the authors considered performing backcrosses to test whether these patterns are influenced by the host genetic background, given that the two lab populations have different origins?

Introduction - Are Istanbul females compatible or incompatible with Slab males? This would be important background information

Lines 98-99 - This observation is the basis of the paper, but there is no data provided and the authors only cite a paper from 2006 in the previous sentence. Please provide or reference the historical data showing that these crosses were compatible from 2005-2017 and provide data on the incompatible crosses observed in 2021. It is unclear if the data in the paper are from this initial observation or if these experiments were done later. 

Line 114 - This section is very brief and lacks context. Is this referring to the data shown in figure 1 or is this a separate experiment? If it's the latter, I would suggest providing the data and also including data prior to 2017 for comparison.

Line 122 - Given that only four individuals were tested from the Ist line is it plausible that there is some undetected variation in cidA and cidB? And if so, could this be contributing to the variability in hatch rates within the compatible crosses in this study? Or perhaps there is additional variation in cidA in Slab females that is contributing to partial CI (as only four compatible individuals were sequenced)?

Line 161 - Is it correct that qPCRs were done in the females only? Given the lack of correlations here it seems likely that this variation in hatch rate is being driven by the male (such higher expression of cidb leading to partial cytoplasmic incompatibility).

Line 214 - The (Β+,16+) female x (β-,16-) male cross has substantially lower and more variable hatch rates than the (β-,16-) Female x (β+,16+) male cross. It's a shame that within-genotype crosses have not been included as a control because this may reflect partial cytoplasmic incompatibility.

Line 216 - In future work, it would be interesting to see how the genotypic frequencies change in the original Slab line given that (β-,16-) is at a frequency of 0% in 2005 and around 25% when the isofemale lines were created. Based on the authors' hypotheses in the discussion it would likely continue to increase.

Line 297 - Since only 4 individuals were tested isn't it possible that variation in cid repertoires in Ist males has gone undetected?

Line 314 - Which dataset is this referring to? I can't see any 2017 data in the paper- please provide it or cite a paper if it has been published previously

Line 320 - While the repertoires must all be at least partially mutually compatible, there is a clear difference in hatch rate between the two reciprocal crosses which may suggest partial incompatibility. There is also no data for within-repertoire crosses so it is unclear if they are fully self-compatible.

Line 439 - Please provide more information about how these populations were maintained prior to experiments. Were they set up as isofemale lines when they were established in 2005, or was this done later?

Line 454 - How did you ensure that mosquitoes were unmated? 

Line 459 - This should be a proportion (number of larvae divided by number of eggs), not a ratio

Line 464 - How is a CI and non-fertilized egg distinguished?

S3 table - for crosses with B-,16- females, were all hatch rates zero?

Reviewer #2:

Cytoplasmic incompatibility (CI) is a remarkable phenomenon found in insects and other arthropods that is caused by bacteria that are transmitted from females to their offspring, by far the most common of which is Wolbachia. In the simplest form of CI, uninfected females produce few or no viable offspring upon mating with infected males, and as a result, infected females have a strong advantage over uninfected ones. There is great interest in using CI microbes to control insect pests and disease vectors and the recent discovery of the genes that cause Wolbachia CI has transformed the field. Wolbachia CI is caused by diverse pairs of genes that appear to function as a toxin (in sperm) and an antidote (in eggs). But we don't yet have a good understanding of how these genes actually work. How does the toxin damage sperm? How does the antidote counteract the toxin? How specific is the interaction between toxin, antidote, and host target? Finally, how do new toxin-antidote combinations evolve?

The authors have been studying these questions in Culex pipiens mosquitoes. Besides its public health importance, Culex is an important model for studying CI because it is infected with diverse Wolbachia strains that exhibit a bewildering combination of compatibility/incompatibility types. Over the past decades, the authors have made a great deal of progress in trying to understand the 'CI code' underlying these combinations by carefully rearing and crossing various isofemale mosquito lines. Here they report that in 2021, they discovered the spontaneous emergence of a new (in)compatibility type in their lab stocks: some females from the 'Slab' line are now incompatible with males from 'Iist' line. They show that this is a result of recombination that has generated a new antidote gene sequence variant. In my opinion, this is an interesting discovery because it shows and captures rapid evolution of Wolbachia CI types. It also helps contribute to our understanding of the logic underlying how CI works. But I have a number of concerns with this manuscript. I hope my comments and suggestions help strengthen the paper and project.

1. I feel that the paper at this stage would be more appropriate for a more specialized audience. As the authors themselves discuss, the generality and ecological relevance of this result is not clear, as the new CI variant is a recently isolated lab mutant. Also, its new incompatibility is cryptic, as it is only exposed in crosses to a different line. To be able to generalize this result, it would be useful and interesting to study the life history of this variant in more detail, as this would help predict its evolution. How does its fitness compare to the ancestral types with which it is co-segregating? Would it persist in 'Slab' experimental populations? 

2. I found the paper to be hard to follow and difficult to read. 

A. The nomenclature/description of CI types/genes is unnecessarily complicated and inaccessible to someone outside of the Wolbachia-Culex field. I fear that these descriptions will scare readers away and I challenge the authors to make this more accessible.

B. Figures 2 and 3 are not clear or visually pleasing. The nomenclature/description is difficult to follow. Also, I find that in these figures, the new lab mutant is lost among all of the other strains that are being considered.

C. The paper (especially the discussion) is too long and hard to follow. Many points of discussion can be shortened. Many of the sections do not need their own sub-headings. The hatching rate variation (and discussion) is confusing. This might be clearer if I had a better sense of what variation looks like when crossing females with males from the same line. The discussion of qPCR variation is confusing. As the authors recognize, it is difficult to disentangle technical artefacts from true variation, but their treatment and discussion of this is not so clear. The discussion of ruling out the possibility of multiple strains of Wolbachia is too long and not so clear. These long and sometimes confusing sections take away from key parts, such as the discussion of how this work connects to theory of CI evolution (i.e. Turelli 1994).

D. The writing can be strengthened for clarity. I found the use of the past tense to be confusing in places, because it is used to describe what was found both in the present study and in previous studies. Below are a few examples/suggestions where the wording is unclear and/or can be tightened up:

-Line 109. Change 'The cidA variant whose presence/absence best matches the CI variations has original aminoacid combinations at its interaction interface with CidB, probably resulting from a recombination.' to 'The cidA variant whose presence best explains the change in compatibility has a novel amino acid sequence at its interaction interface with CidB, probably resulting from a recombination event.' 

-Line 114. Change 'In 2021, crosses between Slab females and Ist males highlighted that some of the Slab females were fully incompatible with Ist males for the first time.' to 'In 2021, we found that some Slab females were fully incompatible with Ist males for the first time.'

-Line 146. I don't understand what 'we metonymically qualified mosquitoes' means. 

Reviewer #3:

This is a nice study from Namais and colleagues who found naturally occurring evolution of their lab colonies and used that to understand the mechanism by which Wolbachia-mediated CI functions and can evolve. This draft appears to be a revision, but I am seeing this for the first time, so I hope my comments are helpful, given this context. My major comments include some suggestions to simply and clarify the text given the density of information, and some requests for more details. Please see below for specific comments.

Line 91 - presumably, the authors mean "in Wolbachia" here?

Line 102-111: The authors mention in the introduction that they ruled out the contamination hypothesis, but I do not see the data directly used to infer this in the results (though I do see mention of this in the discussion). 

Line 114-115: Is there data for this? What proportion of the crosses were compatible? What was the compatibility like prior to this? Not full I presume? Were there intermediate hatch rates here too? It would be great to see the pre- and post- 2017 crosses next to each other in one figure, if possible.

There are quite a lot of abbreviations in the manuscript, and I found I regularly needed to refer back to earlier portions to keep track of which mosquito line was which, the cid variants, crossing directions, etc. For example, the key information that separates categories in figure 1 is in supplemental figure S1, which really interrupts reading/interpretation. Would it be possible to combine some of the figures to better link the genotypes/phenotypes? Or include a schematic that summarizes information from the introduction to show the cid operon with annotations for the important regions (upstream, downstream, cidA-beta,etc), which lines are compatible in which directions, etc? 

Are there statistics for figure 1? (or for any of the other hatching rate comparisons?)

Figure S2: quantification of Wolbachia titers: Is this adult female Wolbachia titer? After they laid eggs? I wonder if this is the most useful sample for determining the relationship between Wolbachia titer and hatch rate. For example, is Wolbachia titer in each embryo more important for whether or not that given embryo will hatch? Obviously some technical challenges when wanting to also determine hatch rate, but perhaps a note that whole-body titers of the mother might not be informative. 

Line 221: To clarify: were all 30 of these mosquitoes from 2005, or was 2005 just the earliest date?

While the molecular methods are relatively standard, it would be best to include the reagents used for PCR/qPCR (e.g., enzyme/mastermix). 

I would like to see more data reported on the nanopore sequencing results, especially because the authors are using coverage information to infer copy number. How many reads were produced for each PCR product? For statements such as line 274 "As these relative coverages were similar", please report the coverage data (total, and by cid). It looks like the method was previously published, but the authors might consider briefly discussing how sequences were compiled and how sequencing errors versus variants were called.

Regarding copy number: are any of the cid operons in putatively functional prophages? Is there a possibility that some phage particles are being made and cid copy number measurements represent both Wolbachia and phage genomes? Or, perhaps a similar processes could be in play if replicative IS elements are copying and moving the regions around.

My final comment is regarding Lines 54-62: While I agree with the authors' assessment that the TA model is a better fit, I do think the discussion of the models (of which there are now arguably three, because lines 40-42 discuss "mod-resc") detracts from the main points. This causes the manuscript to read as fixation on supporting a model, rather than trying to understand the root cause of the very interesting complexity in the wPip system. In general, the introduction is incredibly information dense (alas, by nature of the system), and it just isn't clear what the HM/TA model choice contributes to understanding the data/biology (and indeed, this is not revisited in the discussion). All this being said, I will leave it up to the editor/authors, but my suggestion would simplify, which will make the manuscript appeal to a much broader audience.

---

## [Decision Letter · Decision Letter 2]

21 Dec 2023

Dear Alice,

Thank you for your patience while we considered your revised manuscript "Intra-lineage microevolution of Wolbachia leads to the emergence of new cytoplasmic incompatibility patterns" for publication as a Short Report at PLOS Biology. This revised version of your manuscript has been evaluated by the PLOS Biology editors, the Academic Editor, and two of the original reviewers.

Based on the reviews, we are likely to accept this manuscript for publication, provided you satisfactorily address the remaining points raised by the reviewers and by the Academic Editor (see foot of email). Please also make sure to address the following data and other policy-related requests.

IMPORTANT - please attend to the following:

a) Please address the remaining requests from reviewer #3 and the Academic Editor (note that the latter asks you to move a supplementary Figure into the main paper, and plotting additional existing data in supplementary Figs).

b) Please address my Data Policy requests below; specifically, we need you to supply the numerical values underlying Figs 1, 2C, S1ABCD, S3ABCDEF, S6, S7, either as a supplementary data file or as a permanent DOI’d deposition. I note that you already have an significant number of supplementary Tables; if these contain the data directly underlying the Figs, please clarify and re-name them as supplementary data files; if they do not, please supply the underlying data.

c) Please cite the location of the data clearly in all relevant main and supplementary Figure legends, e.g. “The data underlying this Figure can be found in S1 Data” or “The data underlying this Figure can be found in https://doi.org/10.5281/zenodo.XXXXX”

d) Please make any custom code available, either as a supplementary file or as part of your data deposition.

We expect to receive your revised manuscript within three weeks. 

*Published Peer Review History*

*Press*

Sincerely,

Roli

Roland Roberts, PhD

Senior Editor,

rroberts@plos.org,

PLOS Biology

DATA POLICY:

Regardless of the method selected, please ensure that you provide the individual numerical values that underlie the summary data displayed in the following figure panels as they are essential for readers to assess your analysis and to reproduce it: Figs 1, 2C, S1ABCD, S3ABCDEF, S6, S7. NOTE: the numerical data provided should include all replicates AND the way in which the plotted mean and errors were derived (it should not present only the mean/average values).

CODE POLICY

Per journal policy, as the code that you have generated is important to support the conclusions of your manuscript, we require that you make it available without restrictions upon publication. Please ensure that the code is sufficiently well documented and reusable, and that your Data Statement in the Editorial Manager submission system accurately describes where your code can be found.

DATA NOT SHOWN?

REVIEWERS' COMMENTS:

Reviewer #1:

The authors have addressed all my previous suggestions which has led to a substantially improved manuscript. I have no other major comments to make. 

Reviewer #3:

Thanks to the authors for a careful and thorough revision of their manuscript. I maintain that this is a really interesting finding and the authors have done a lot of nice work to characterize the mechanisms responsible for these recently evolved CI patterns. Please see below for some minor comments to help with clarity. 

Line 23: "In wPip, CI is thought to function as a toxin-antidote (TA) system, with cidA

the antidotes, cidB the toxins and compatibility relying on having the right antidotes in the female to bind and neutralize the male's toxins" -- this grammar reads oddly to me. Suggest: 

"In wPip, CI is thought to function as a toxin-antidote (TA) system where compatibility relies on having the right antidotes (CidA) in the female to bind and neutralize the male's toxins (CidB)".

Line 29 "original sequence at its binding interface, matching the original sequence at the toxin's binding" - is it the sequence that truly matches? Or just the shape of the protein?

I think there is still an opportunity to clarify the report of the initial key finding of incompatibility (lines 110-116). The sentences indicate 2021 data showing newly formed incompatibility (my assumption is that these have not yet been published, and the crosses were performed in 2021 - but I think the confusion is because the key Sicard et al paper was published in 2021). There are no data presented to accompany these details, and the reader has to infer the differences since 2017. I think it would be so much more impactful to re-plot the 2017 crosses next to the 2021 crosses. Finally, Line 116 "The hatching rates of the 69 compatible clutches varied between from 3.2% to 96.4%" - which crosses here are "compatible"? Or, these are the clutches from the newly incompatible cross in which at least some larvae hatched? I think it is confusing to call them compatible when the argument is that this cross is no longer compatible. If I am correct in my interpretation here, I suggest:

In the present study (crosses performed in 2021), we witnessed for the first time the emergence of full incompatibly between some Slab females and Ist males. Twenty five of 94 rafts had a 0% hatch rate, indicating strong CI, and the remaining 69 clutches had hatch rates between 3.2% to 96.4%.

Line 142: "specifically screened the 89 females" - the previous section discussing the females 102 females, 94 of which were mated. It is unclear what the 89 is referring to.

Line 147: I agree that with the other reviewer that the phrase "metonymically qualified mosquitoes" is jargon - I also agree with the authors response that it makes sense to do this, but perhaps consider editing the manuscript to instead state "hereinafter we refer to a mosquitos by its Wolbachia infections' cidA genotype repertoire (e.g., (β+,16+), (β+,16-), or (β-,16-)"

Line 310 "Our microevolution approach is another functional strategy, additional to transgene expression…" - I agree with the conclusion that the data support T/A but, it does seem unusual to call it "your approach", as the authors were more very lucky and very observant to catch this phenomenon… rather than intentionally evolving lines experimentally. 

Some of the figures use a combination of red and green, which is likely not colorblind friendly.

COMMENTS FROM THE ACADEMIC EDITOR:

I would add that in line 123 the reference should be to another figure or table, not S1 Fig.

Also, I think that figure S2 should be included in figure 2. This is the clearest visualization of the relationship between recombination, sequence, and compatibility phenotypes.

Finally, the data on hatching rates present in many of the tables in supplementary information should be represented in supplementary figures too.

---

## [Editor Report · Decision Letter 3]

8 Jan 2024

Dear Alice,

Thank you for the submission of your revised Short Report "Intra-lineage microevolution of Wolbachia leads to the emergence of new cytoplasmic incompatibility patterns" for publication in PLOS Biology. On behalf of my colleagues and the Academic Editor, Luis Teixeira, I'm pleased to say that we can in principle accept your manuscript for publication, provided you address any remaining formatting and reporting issues. These will be detailed in an email you should receive within 2-3 business days from our colleagues in the journal operations team; no action is required from you until then. Please note that we will not be able to formally accept your manuscript and schedule it for publication until you have completed any requested changes.

IMPORTANT: I've asked my colleagues to include the following editorial request: "Many thanks for citing the location of the data in the legends to the Supplementary Figures. Please could you do the same in the legends for the main Figs 1 and 2C, e.g. 'The data underlying this Figure can be found in S1 Data'” 

Sincerely,

Roli

Senior Editor

PLOS Biology

rroberts@plos.org